# Nonlinearity of the post-spinel transition and its expression in slabs and plumes worldwide

Junjie Dong [1,2,6] ✉, Rebecca A. Fischer [1], Lars P. Stixrude [3], Matthew C. Brennan [1,7], Kierstin Daviau[1,8], Terry-Ann Suer[1,9], Katlyn M. Turner[1,10], Yue Meng[4] & Vitali B. Prakapenka [5]

Phase transitions in the mantle control its internal dynamics and structure. The post-spinel transition marks the upper–lower mantle boundary, where ringwoodite dissociates into bridgmanite plus ferropericlase, and its Clapeyron slope regulates mantle flow across it. This interaction has previously been assumed to have no lateral spatial variations, based on the assumption of a linear post-spinel boundary in pressure and temperature. Here we present laser-heated diamond anvil cell experiments with synchrotron X-ray diffraction to better constrain this boundary, especially at higher temperatures. Combining our data with results from the literature, and using a global analysis based on machine learning, we find a pronounced nonlinearity in the post-spinel boundary, with its slope ranging from −4 MPa/K at 2100 K, to −2 MPa/K at 1950 K, and to 0 MPa/K at 1600 K. Changes in temperature over time and space can therefore cause the post-spinel transition to have variable effects on mantle convection and the movement of subducting slabs and upwelling plumes.

High-pressure phase transitions in mantle minerals cause seismic discontinuities and influence convection in the Earth's interior. One such transition is the pressure-induced dissociation of ringwoodite (rw, $(Mg,Fe)_2SiO_4$) into bridgmanite plus ferropericlase (bm + fp, $(Mg,Fe)SiO_3 + (Mg,Fe)O$)[1–4]. Often referred to as the post-spinel transition, it corresponds to a global seismic discontinuity in the mantle at a depth of about 660 km[5]. At this depth, the mantle experiences significant changes in temperature, material flow, and buoyancy, the effects of which are largely controlled by the Clapeyron slope of the post-spinel transition ($\gamma_{post-spinel} = \frac{\partial P}{\partial T} = \frac{\Delta S_{post-spinel}}{\Delta V_{post-spinel}}$, where $\Delta S_{post-spinel}$ and $\Delta V_{post-spinel}$ are the entropy and volume changes across the transition, respectively)[6–8]. For example, if $\gamma_{post-spinel}$ is more negative, this

transition can cause subducting slabs to stagnate at mid-mantle depths[9,10], and it can also impede the upwelling of mantle plumes[11,12].

The post-spinel transition between rw and bm + periclase (pe) in the iron-free system ($Mg_2SiO_4 \leftrightarrow MgSiO_3 + MgO$) is useful for understanding this transition in the Earth's pyrolitic mantle, since the post-spinel Clapeyron slopes in these two compositions are nearly identical[1–4,7,13] (Figure S1). Numerous experimental studies have directly investigated the phase stability of rw and bm + pe using in situ X-ray diffraction[14–19], but with large variations in their derived Clapeyron slopes ranging from about −3 MPa/K to +1 MPa/K. These previous studies typically assume a linear boundary due to the large amount of scatter in the data. This is equivalent to assuming that upwelling or downwelling

[1]Department of Earth and Planetary Sciences, Harvard University, Cambridge, Massachusetts, USA. [2]Department of the History of Science, Harvard University, Cambridge, Massachusetts, USA. [3]Department of Earth, Planetary, and Space Sciences, University of California, Los Angeles, California, USA. [4]High Pressure Collaborative Access Team (HPCAT), X-Ray Science Division, Argonne National Laboratory, Argonne, IL, USA. [5]Center for Advanced Radiation Sources, University of Chicago, Chicago, IL, USA. [6]Present address: Now at Division of Geological and Planetary Sciences, California Institute of Technology, Pasadena, California, USA. [7]Present address: Now at Shock and Detonation Physics Group, Los Alamos National Laboratory, Los Alamos, NM, USA. [8]Present address: Now at Toi-Ohomai Institute of Technology, Tauranga, New Zealand and School of Science, University of Waikato, Tauranga, New Zealand. [9]Present address: Now at Laboratory for Laser Energetics, University of Rochester, Rochester, New York, USA. [10]Present address: Now at MIT Media Lab, Massachusetts Institute of Technology, Cambridge, Massachusetts, USA. ✉e-mail: dong2j@caltech.edu

mantle flow across this boundary will experience the same Clapeyron slope, regardless of temperature variations in space and time. In fact, thermodynamic models have suggested that the post-spinel boundary may be nonlinear, with a slope that varies with temperature[6,20,21]. If the boundary is nonlinear, temperature differences (for example, between the present-day and early Earth, or between slabs and plumes) could lead to different effects of the same post-spinel transition on mantle flow due to local differences in the Clapeyron slope. However, the empirical thermodynamic models rely on experimental data collected at low temperatures, such as heat capacity (< 500 K)[20,21], which, when extrapolated, produce inaccurate entropy and volume at the high temperatures relevant to the mantle transition zone (1600–2200 K), thus compromising their reliability in predicting the Clapeyron slope. Chanyshev et al.[19] present the first experimental evidence that the post-spinel boundary is nonlinear, but do not quantify the variation in its Clapeyron slope. To date, there is no robust estimate of the nonlinearity of the post-spinel transition.

To understand the effects of the nonlinear post-spinel transition on mantle convection, we have placed new experimental constraints on the Clapeyron slope of this transition and clarified how this slope changes with temperature. Previous efforts to determine this slope from direct observations of phase stability have faced challenges due to: 1) relying on visual inspection and freehand drawing of the boundary, which can introduce biases based on the researchers' expectations; 2) using individual datasets rather than combining them,

which prevents evaluation of the accuracy of each dataset; and 3) obtaining in situ observations on $Mg_2SiO_4$ primarily only at temperatures of 1500–2000 K, which leads to errors when extrapolating the phase boundary to higher temperatures.

To address these challenges, we have performed laser-heated diamond anvil cell (LH-DAC) experiments using synchrotron X-ray diffraction (XRD) on $Mg_2SiO_4$ across a range of pressures ($P$) and temperatures ($T$) relevant to the Earth's mantle transition zone (16–28 GPa, 1573–2723 K). Our LH-DAC data span multiple phase boundaries in $Mg_2SiO_4$, with a particular focus on the previously-understudied triple point of wd–rw–bm+pe. Combining our dataset including the triple point with previous high-precision multi-anvil (MA) press datasets, we used logistic regression and supervised learning to identify the phase boundaries in $Mg_2SiO_4$ under mantle transition zone conditions and determine the nonlinearity of the post-spinel boundary. Finally, we modeled the lateral variations in the post-spinel Clapeyron slope at the base of the transition zone due to variations in temperature, and evaluated its impact on slabs, plumes, and the ambient mantle.

## Results and discussion

### In situ experimental observations on the $Mg_2SiO_4$ phase diagram at transition zone conditions

We present an experimental dataset showing high $P$–$T$ phase stability observations of $Mg_2SiO_4$. These data were obtained using synchrotron

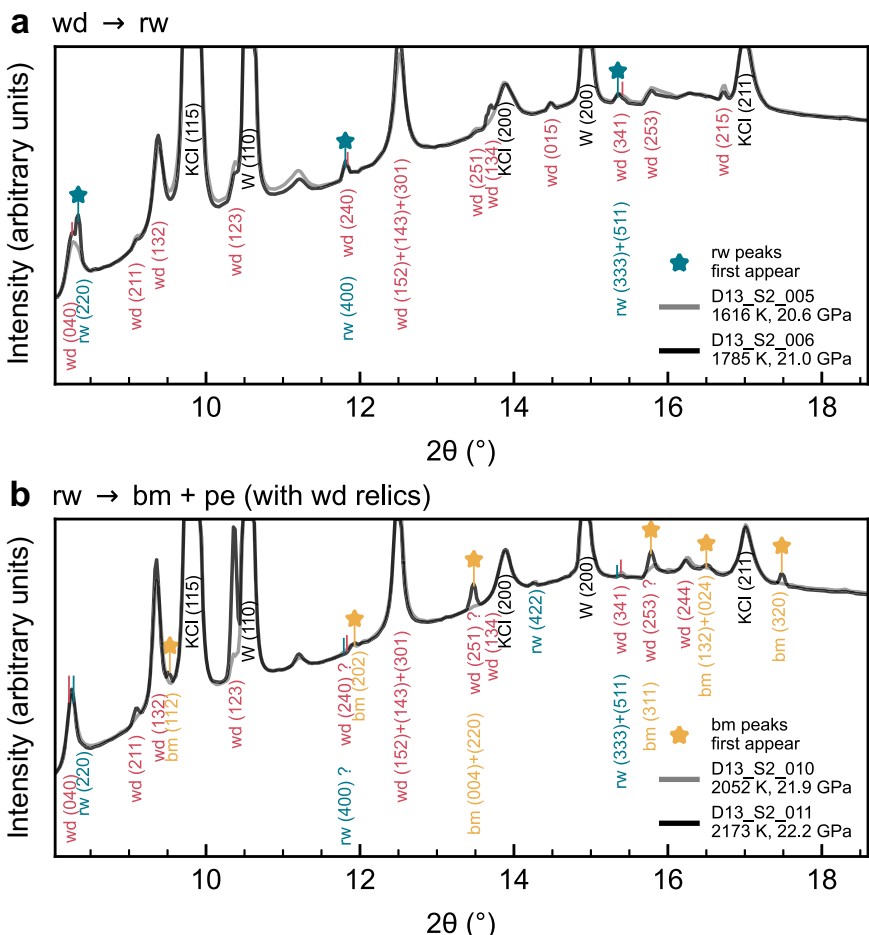

**Fig. 1 | In situ phase stability observations in $Mg_2SiO_4$ collected in LH-DAC experiments with synchrotron XRD.** Representative XRD patterns from one heating cycle are shown, demonstrating the first appearance of the peaks of a new phase (marked by colored stars). **a** All visible diffraction peaks at 1616 ± 162 K and 20.6 ± 1.3 GPa can be attributed to wadsleyite (wd), and wd is the only stable phase of $Mg_2SiO_4$. Upon heating to 1785 ± 187 K and 21.0 ± 1.5 GPa, ringwoodite (rw) peaks appear for the first time. **b** By 2052 ± 218 K and 21.9 ± 1.7 GPa, the rw peaks have continued to grow, with some possibly overlapped by relic wd peaks. After heating further to 2173 ± 252 K and 22.2 ± 1.9 GPa, the bridgmanite (bm) peaks appear for the first time. All peaks are labeled with Miller indices (*hkl*).

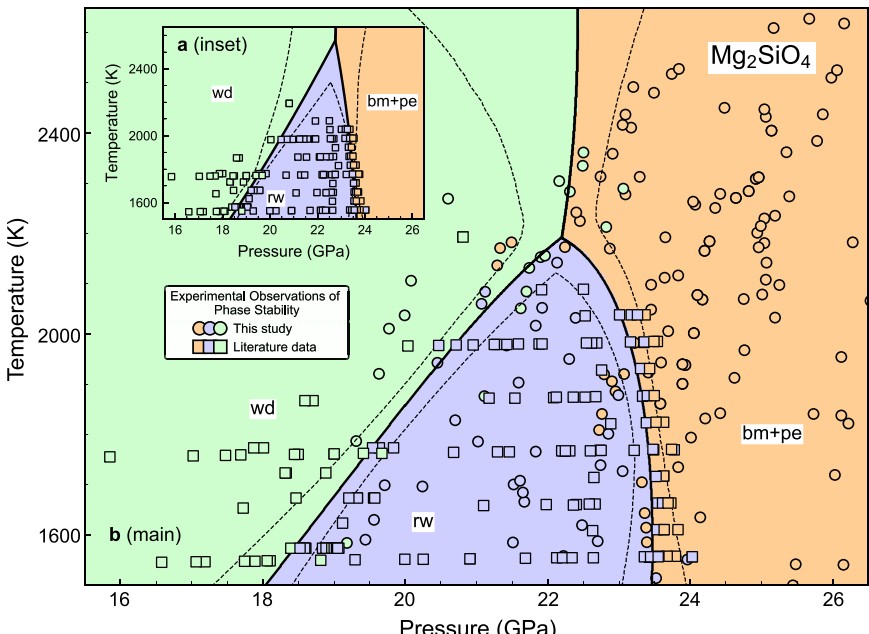

**Fig. 2 | Global analyses of the Mg₂SiO₄ phase diagram at mantle transition zone conditions.** Stability fields of wd, rw and bm + pe are predicted between 1500 and 2700 K and between 15 and 27 GPa with a 95% confidence interval (CI) (dashed lines). For the initial analysis (**a, inset**), we used only MA literature data. For the final analysis (**b, main**), we combined these literature data with our new LH-DAC dataset. In situ experimental observations are shown as circles (this study, LH-DAC) and squares (four literature datasets, MA[19,22–24]). Phases are color-coded as follow: wd in green, rw in purple, and bm + pe in orange.

XRD in nine LH-DAC experiments, which included 17 laser-heated spots (Supplementary Data 1; Figure S2). Figure 1 shows examples of high $P-T$ XRD patterns that constrain phase boundaries in Mg₂SiO₄. In Fig. 1a, the wd ↔ rw transition is highlighted. Initially, the wd phase remained stable at 1616 ± 162 K and 20.6 ± 1.3 GPa. However, in the subsequent diffraction pattern collected at 1785 ± 187 K and 21.0 ± 1.5 GPa, a transition to the rw phase began. Figure 1b focuses on the post-spinel transition, where the rw peaks initially continue to grow and increasingly overlap with the relic wd peaks. In the following XRD pattern, bm peaks appear for the first time, with the rw phase beginning its transition to bm + pe at 2173 ± 252 K and 22.2 ± 1.9 GPa.

To combine our results with literature datasets and perform a global analysis of the Mg₂SiO₄ phase diagram, we have compiled all available in situ phase stability data on the Mg₂SiO₄ system from previous studies (Supplementary Data 2). Here we utilize a selected compilation of four specific data sets[19,22–24] (Fig. 2). These data were all obtained from in situ high-precision MA experiments using an MgO pressure calibrant, and we recalculated their pressures so that all data in the compilation are based on internally-consistent pressure calibrations[25,26]. We also corrected the thermocouple-based temperature measurements of these studies for the effect of pressure on electromotive force[27,28] to maximize consistency with the spectro-radiometric temperature measurements in this study.

## Global inversion of the post-spinel phase boundary and its degree of nonlinearity

After correcting and combining the four MA datasets from the literature[19,22–24], we globally inverted them into a $P-T$ phase diagram of Mg₂SiO₄, first excluding our LH-DAC dataset (Fig. 2a, inset). These literature data were all obtained at temperatures below ~2100 K. This initial analysis produces linear boundaries for both wd ↔ rw and the post-spinel transition. However, the limited temperature range of this MA dataset creates challenges in accurately determining the $P-T$ conditions of the triple point. For example, the initial global analysis suggested a triple point involving wd, rw, and bm + pe above 2600 K, which directly contradicts all existing experimental and theoretical

evidence[15,29–32]. This discrepancy highlights the unreliability of extrapolated phase boundaries and emphasizes the need to combine MA and LH-DAC data for phase boundary determination over a wider temperature range.

Our LH-DAC dataset fills this experimental gap, providing abundant in situ phase stability observations for wd, rw, and bm + pe over a temperature range of 1573–2723 K at mantle transition zone pressures (Figure S3). In our second and final global inversion (Fig. 2b, main), incorporating our latest LH-DAC data with the four literature MA datasets[19,22–24], we found a nonlinear Clapeyron slope for the post-spinel transition. Specifically, the slope changes from $-4.1^{+2.1}_{-7.6}$ MPa/K at 2150 ± 50 K to $-1.7^{+1.1}_{-1.4}$ MPa/K at 1900 ± 50 K and finally to $0.0^{+1.2}_{-1.7}$ MPa/K at 1600 ± 50 K. The wd ↔ rw and wd ↔ bm + pe transitions were found to be nearly linear with slopes of $5.7^{+1.7}_{-1.0}$ MPa/K at 1800 ± 200 K and $0.8^{+2.1}_{-10.3}$ MPa/K at 2300 ± 200 K, respectively. We estimated the location of the triple point to be ~22.2 GPa and ~2190 K.

The post-spinel boundary is remarkably nonlinear, in contrast to the wd ↔ rw boundary over a similar temperature range. This clear difference is consistent with the thermodynamic properties of the Mg₂SiO₄ polymorphs. The Clapeyron slope of a phase transition depends on ΔS and ΔV across the transition, $\gamma = \frac{\Delta S}{\Delta V}$. =. Both ΔS and ΔV are influenced by the thermal expansivities (α) of the phases ΔV involved. For the wd ↔ rw transition, $\alpha_{rw}$ and $\alpha_{wd}$ show similar temperature dependencies. However, for the post-spinel transition, $\alpha_{rw}$ increases more slowly than $\alpha_{bm}$ at high temperatures[32,33]. This discrepancy in the temperature dependencies of these thermal expansivities means that $\Delta V_{wd↔rw}$ and thus also $\gamma_{wd↔rw}$ varies less than $\Delta V_{post-spinel}$ and $\gamma_{post-spinel}$ with changes in temperature. Consequently, the thermodynamic properties of these minerals are consistent with the findings of our global inversion, with a boundary that appears linear for the wd ↔ rw transition and clearly nonlinear for the post-spinel transition.

Our LH-DAC dataset covers a wider temperature range that extends beyond the triple point of wd–rw–bm+pe, which is critical for constraining the nonlinearity of the post-spinel boundary and is only possible in the LH-DAC. However, using the LH-DAC to achieve these

higher temperatures comes with the trade-off of lower precision (both in pressure and temperature) than in a multi-anvil press[19]. Here, we combine our LH-DAC data with the selected MA data from the literature, to take advantage of the higher-precision constraints on the phase boundary at lower temperatures obtained in those MA studies while significantly improving constraints near the triple point, and thus the nonlinearity of the post-spinal boundary, through the addition of the higher temperature LH-DAC data. Future work should aim to probe closer to the post-spinel boundary with improved experimental setups that balance achieving high temperatures and measurement precision.

## Mantle expression of the nonlinear post-spinel boundary in slabs and plumes worldwide

Solid–solid phase transitions shape mantle convection. While a small subset of cold slabs and hot plumes are likely influenced by the akimotoite (ak) ↔ bm and post-garnet transitions[4,19,34], the dynamics of the Earth's mantle today, especially for most slabs and plumes, are primarily controlled by the post-spinel transition[1–4]. When the post-spinel boundary has a more negative Clapeyron slope, it impedes mantle flow more strongly[7]. The nonlinear nature of this boundary indicates that the magnitude of its negative slope, and hence its impedance to mantle flow, varies with temperature. To understand how such nonlinearity manifests itself in the mantle, we constructed a temperature map at the "660 km" discontinuity (labeled T660; Fig. 3a, b). We then examined variations in its Clapeyron slope ($\gamma_{post-spinel}$) in slabs, plumes, and ambient mantle at this depth (Fig. 3c).

For slabs undergoing the post-spinel transition (T660 ≥ 1400 K, 46 of the 50 slabs we studied[35]; Supplementary Data 4), the average T660 is 1600 ± 130 K (standard deviations applied throughout), and the average $\gamma_{post-spinel}$ is −0.2 ± 0.4 MPa/K. In contrast, plumes associated with the post-spinel transition (T660 < 2150 K, 18 of the 26 plumes[36] we studied; Supplementary Data 5) have an average T660 of 2060 ± 80 K and a $\gamma_{post-spinel}$ of −3.6 ± 1.4 MPa/K. For plumes, the average slope is eighteen times more negative than its value for slabs. Such a strong contrast in their post-spinel Clapeyron slopes would result in a significantly different impedance to flow for subducting slabs and rising plumes depending on their locations (Fig. 3).

Half of the subducting slabs in Earth's mantle today are relatively cold, with T660 = 1400–1600 K, most of which are clustered in the western Pacific region. These slabs have a Clapeyron slope of ~0 MPa/K. As a result, they would encounter minimal impedance to subduction at 660 km. In contrast, warmer plates near regions such as southern Chile (with T660 = 1760 K), Colombia/Ecuador (with T660 = 1830 K), and northern Cascadia (T660 = 1940 K) have more negative slopes ranging from −0.7 MPa/K to −1 MPa/K and −1.6 MPa/K, respectively. These warmer slabs are therefore expected to encounter greater resistance during subduction at the post-spinel transition.

Many mantle plumes associated with the post-spinel transition today (T660 < 2150 K) would exhibit different dynamic behaviors and morphologies[11], consistent with its nonlinear slope. Some of these plume heads (e.g., St. Helena, Canary, and Reunion) tend to be trapped at around 660 km depth with $\gamma_{post-spinel}$ more negative than about −3.4 MPa/K, while others (e.g., Ascension, Cameroon, and Cape Verde) continue to rise through the transition zone with $\gamma_{post-spinel}$ more positive than about −2.8 MPa/K. In the range of $\gamma_{post-spinel}$ between −3.4 and −2.8 MPa/K (e.g., Azores and Kerguelen/Heard), ring-shaped secondary plumes can form. In addition, the dynamic topography of these plumes can also be influenced by the nonlinear post-spinel boundary, causing the surface subsidence to decrease with a more positive $\gamma_{post-spinel}$[37]: from 100 m with $\gamma_{post-spinel}$ = −3.0 MPa/K at T660 = 2055 K, to 200 m with $\gamma_{post-spinel}$ = −3.2 MPa/K at T660 = 2065 K.

On one hand, this nonlinear post-spinel boundary has implications for the evolution of mantle convection from the early Earth to the present. During the Archean, slabs, plumes, and the ambient mantle may have been 150–200 K hotter than today[38,39]. As a result, these slabs, plumes, and ambient mantle would have had more negative post-spinel slopes compared to their present-day equivalents. These substantial differences could have led to a form of dynamic layering within the early mantle that differed from the whole-mantle convection of today[7,40]. On the other hand, this nonlinearity in the post-spinel boundary also affects the thermal evolution of individual stagnant slabs and plumes at any given time. The variable effects of rheology and buoyancy, driven by the temperature-dependent post-spinel Clapeyron slope, could significantly influence their dynamics and shapes. This influence could persist throughout their lifetimes until they reach thermal equilibrium with the ambient mantle.

Our global map of lateral variations in the post-spinel Clapeyron slope (Fig. 3f) provides a spatial guide for studying slabs and plumes worldwide, each with its specific Clapeyron slope. Furthermore, the nonlinear post-spinel boundary introduces varying dynamic behaviors of the mantle through time. This variability is intrinsically linked to mantle temperature differences across time and space, and is largely unexplored in existing geodynamical models. Mantle flow is influenced by several factors[10], including viscosity and compositional differences in the mantle, and the configuration and history of the specific slab and plume, in addition to the Clapeyron slopes of major phase transitions, so it is not always straightforward to predict slab or plume behavior at 660 km depth from only the post-spinel Clapeyron slope. Therefore, it is imperative for future studies to accurately model the thermal and dynamical evolution of slabs, plumes, and the ambient mantle consistent with a temperature-dependent post-spinel Clapeyron slope to fully understand mantle dynamics and its temporal and spatial variability.

## Methods
### Synchrotron x-ray diffraction experiments
Short symmetric diamond anvil cells and a gas membrane cell were used to generate high pressure conditions. Double-sided laser heating and synchrotron XRD measurements were performed at beamline 13-ID-D (GeoSoilEnviro Center for Advanced Radiation Sources, GSECARS)[41,42] and beamline 16-ID-B (High Pressure Collaborative Access Team, HPCAT)[43] at the Advanced Photon Source (APS), Argonne National Laboratory (ANL). The starting materials were powdered synthetic forsterite ($Mg_2SiO_4$) mixed with tungsten (W), sandwiched between two layers of potassium chloride (KCl) and loaded into a sample chamber predrilled into a rhenium (Re) gasket. The KCl layers served as the pressure medium, thermal insulator, and primary pressure standard[26,44], while the W was used as a laser absorber and secondary pressure standard[25]. Both W and KCl have melting temperatures higher than $Mg_2SiO_4$[45,46] and are chemically inert in this pressure and temperature range. The P–T path of each experiment is shown in Figure S2. Further details of our experimental procedures and results are given in the Supplementary Information.

### Phase identification and phase boundary detection
We adopted two main criteria to detect phase transitions. First, we observed the emergence of the first peaks from any new phase between two consecutive diffraction patterns. Second, we monitored the peak intensity of existing phases in each heating cycle. However, the intensities of these peaks can change due to the preferred orientations of new crystal grains or peak sharpening during heating[19]. As an added measure, we used the "spottiness" of the Debye rings in 2D XRD images to track the stability of existing phases when the intensities of their diffraction peaks do not change uniformly. It is important to distinguish actual phase transitions from changes in peak intensity and d-spacing caused by temperature fluctuations. To avoid misinterpretation, we did not use any diffraction patterns in which the temperature fell by over 50 K during heating.

The main goal of these LH-DAC experiments was to investigate the location of the $Mg_2SiO_4$ triple point and obtain phase stability data in

the high-temperature region between the triple point and the solidus. To achieve these goals, we began our experiments at pressures 3–4 GPa below our target range to accommodate thermal pressure upon heating. We initially compressed and heated our samples to 17–20 GPa and 1300–1600 K, then to our target range of 21–23 GPa and 1900–2300 K. Since these *P–T* conditions are so close to the wd–rw

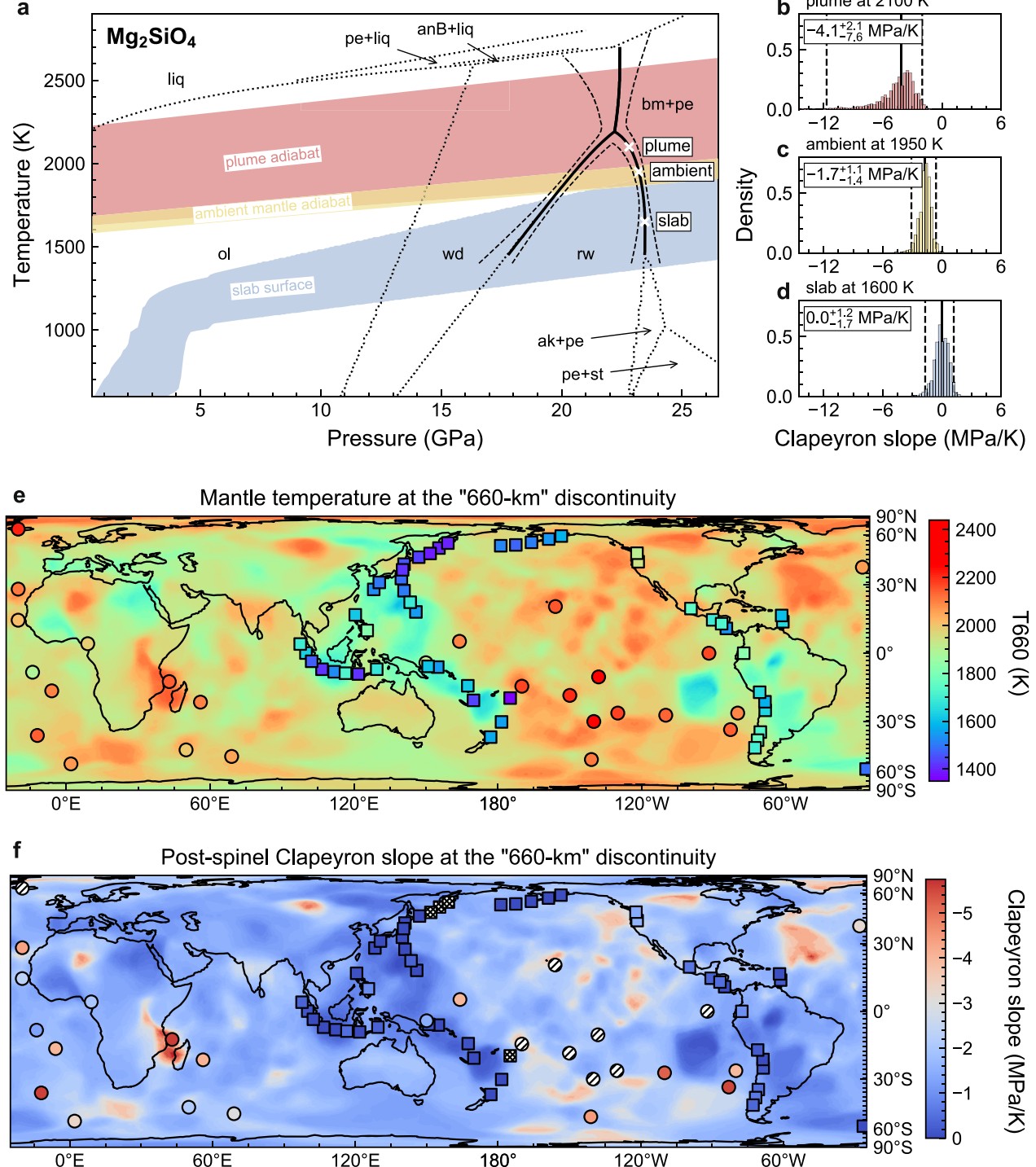

**Fig. 3 | Nonlinearity of the post-spinel transition and its expression in slabs and plumes worldwide. a** Phase diagram for $Mg_2SiO_4$ up to 26 GPa, showcasing three phase boundaries: wd ↔ rw, wd ↔ bm + pe, and rw ↔ bm + pe at the Earth's mantle transition zone conditions. Solid black lines indicate the phase boundaries that have been statistically optimized in this study. The melting boundaries and the ol ↔ wd, ak ↔ bm, and pe + stishovite (st) ↔ bm boundaries (dashed black lines) are based on Li and Stixrude and Lithgow-Bertelloni[4,62]. **b–d** Post-spinel Clapeyron slopes at 2100 K (plume, **b**), 1950 K (ambient mantle, **c**), and 1600 K (slab, **d**), with 95% CI (dashed lines), obtained using our machine learning method. **e–f** Temperature and post-spinel Clapeyron slopes at the "660-km" seismic discontinuity around the world (Supplementary Data 4–5). Subducting slabs (squares) and plume-fed hotspots (circles) are color-coded based on their T660s and post-spinel Clapeyron slopes. Additionally, those primarily influenced by the ak ↔ bm transition (occurring at T660 < 1400 K in slabs) and the post-garnet transition (occurring at T660 ≥ 2150 K in plumes) are distinguished by hatched lines and crosses, respectively.

phase boundary[19], we have consequently often observed the new rw phase and the metastable wd phase coexisting. Detection of the initial rw peaks was challenging due to the high diffraction peak density of the orthorhombic wd structure.

Although it would be ideal to reverse the $P-T$ path and observe these phase transitions upon cooling, this proves challenging in LH-DAC experiments. Completely recombining bm + pe back into a single phase within the fast-paced experimental timeframe is difficult due to the slow diffusion rate[47] and the spatial separation of the bm and pe grains[48]. Reversal experiments are generally only feasible in MA experiments, where the compression–decompression and heating–cooling cycles can be controlled simultaneously and incrementally. However, these MA reversal experiments have limitations, including the accuracy of the temperature measurements, which are affected by thermocouple calibrations and pressure effects on electromotive force (emf) at temperatures over ~1000 K. These can lead to significant systematic errors in temperature measurements, causing, for example, a > 80 K error at 1773 K with a type D thermocouple, which impacts the determination of Clapeyron slopes[27]. Hence, we found LH-DAC experiments effective for collecting a larger dataset with smaller systematic errors, despite the potential for greater random errors in temperature measurements.

## Selection and correction of phase stability observations from the literature

We performed a global analysis of the $Mg_2SiO_4$ phase diagram by incorporating our LH-DAC dataset with selected MA datasets from the literature. We used four key datasets from Katsura et al. [22], Inoue et al. [23], Katsura et al. [24], and Chanyshev et al. [19]. All of these data are from in situ MA experiments with MgO as the primary pressure scale. The pressures from these datasets were recalculated using an MgO pressure scale that was cross-calibrated against KCl by Sokolova et al. and Tateno et al. [25,26]. The rest of the existing experimental datasets on $Mg_2SiO_4$ from the literature can be found in Supplementary Data 2–3, but they were omitted from our global analysis for one or more of the following reasons (Figure S4):

- **Ex situ experiments:** Many earlier experimental studies determined phase boundaries using ex-situ methods. MA examples include works by Ito and Takahashi [1] and Ishii et al.[49], in which pressures were based on "fixed point" calibrations (typically with one calibration at room temperature and one at high temperature). These experiments are expected to have variable pressures with changing temperature, influenced by factors like thermal pressure and material relaxation[50]. DAC examples include Chudinovskikh and Boehler[29], in which the pressures were based on estimated thermal pressures from past experiments. Comparing and correcting the pressures from these two type of ex situ datasets is impossible, and hence, they were excluded.

- **In situ experiments with Au and/or Pt pressure scales:** Notable differences in phase boundaries are found among literature datasets with different pressure scales (MgO, Au, Pt, etc.). Ye et al. observed significant systematic bias between MgO, Au, and Pt pressure scales, even after recalibrating them[51]. This phenomenon may be due to severe anharmonic effects in Au and a large deviation from Debye-like behavior in Pt at high temperatures. To maintain maximum internal consistency, datasets using Au or Pt pressure scales, including Irifune et al., Shim et al., Kastura et al., and Ghosh et al.[16,18,50,52], were excluded.

- **In situ experiments with uncorrectable pressures and/or temperatures**: Some in situ experimental studies, such as Fei et al.[17], used a type C thermocouple (W95Re5–W74Re26) that lacks an extrapolatable calibration for its pressure effects. As a result, these datasets were also excluded.

## Multi-class logistic regression and supervised learning

Determining phase boundary locations can be considered as a classification problem[53–55]: assigning observations to a specific stable mineral phase under certain $P-T$ conditions. We used multi-class logistic regression here for determining the phase diagram of $Mg_2SiO_4$. This approach can accommodate multiple stable phases and gives probabilities of observing these phases at specific $P-T$ conditions, $\hat{p}(Y = k|P,T)$, which can be expressed as a logistic function, $\frac{e^{\sum_{i,j=0}^{n}\beta_{i,j}^k P^i T^j}}{1+e^{\sum_{i,j=0}^{n}\beta_{i,j}^k P^i T^j}}$. The log-odds, $\ln\frac{\hat{p}(Y=k|P,T)}{1-\hat{p}(Y=k|P,T)}$, can be interpreted as a polynomial function of $P$ and $T$:

$$\ln\frac{\hat{p}(Y=k|P,T)}{1-\hat{p}(Y=k|P,T)} = \sum_{i,j=0}^{n} \beta_{i,j}^k P^i T^j = f(P,T) \tag{1}$$

We can then convert probability estimates from three separate models (with $k$ = wd, rw, or bm + pe; $K$ = 3) to one set of probability estimates using the "softmax" function[53,55]. The rescaled probability estimates add up to 1. We take the stable phase to be the class with the highest probability, and the tripe point occurs where $\hat{p}(Y = \text{wd}) = \hat{p}(Y = \text{rw}) = \hat{p}(Y = \text{bm + pe}) = \frac{1}{3}$. The coefficients $\beta_{i,j}^k$ are estimated by minimizing a combined negative log-likelihood function, or total cross entropy, $-L$[53]:

$$-L = -\frac{1}{M}\sum_{m=1}^{M}\sum_{K=1}^{K}\{t_{m,k}(y_m = k)\cdot\ln[P_m(y_m = k)] \\ + t_{m,l}(y_i \neq k)\cdot\ln[1 - p_m(y_m \neq k)]\} \tag{2}$$

where $t_{m,l}(y_i = k)$ is 1 if and only if the observation $m$ belongs to phase $k$, and $p_m(y_m = k)$ is the output probability that the observation $m$ belongs to phase $k$.

We then used supervised learning (Figure S5) to constrain this multi-class logistic model using the experimental phase stability observations on $Mg_2SiO_4$ from this study and from the literature. We first divided the compiled data, with 70% being a train set and 30% being a test set. Evaluating the model on both of these sets was crucial to avoid overfitting because the model may overfit the train set to predict perfect responses but fail to perform well on the unseen test set. Our approach was to describe the log-odds using polynomials of P and T from degree 1 to 10 ($n$ = 1–10) in Eq. 1. However, overfitting can occur if a high degree polynomial for the log-odds is used, and thus we applied a regularization method, called Lasso or L1 regularization. This regularization method constrains or regularizes the coefficient estimates, $\beta_j$, by modifying the least-squares loss function in Eq. 2, $L(\beta)$, into a regularized loss function, $L_{\text{Lasso}}(\beta) = L(\beta) + \lambda\sum_{i,j=1}^{n}|\beta_{i,j}|$, where $\lambda$ is a scaler that assigns weights to the regularization term, or the regularization strength, $\lambda\sum_{i,j=1}^{n}|\beta_{i,j}|$. We then used $\lambda$ to discourage/penalize extreme values of $\beta_j$ to avoid overfitting: when $\lambda$ is sufficiently large, the regularized loss function $L_{\text{Lasso}}(\beta)$ becomes increasingly sensitive to $\lambda\sum_{i,j=1}^{n}|\beta_{i,j}|$; in such a scenario, a successful convergence would shrink $\beta_{\text{Lasso}}$ to zero, or close to zero.

We optimized the hyperparameter $C$ (the inverse of regularization strength, $\lambda = \frac{1}{C}$), along with other parameters in the "scikit-learn" package such as "multi_class" and "solver"[56]. Parameter tuning was achieved through a grid search with k-fold cross-validation, which involved further splitting the train set into k folds ($k$ = 3–5). We then trained the model using $k$–1 folds, validating it on the remaining fold. The final evaluation of the parameters was based on the average over the $k$ folds.

For model evaluation, we used a combination of two metrics: precision ($\frac{\text{true positive}}{\text{true positive + false positive}}$) and recall ($\frac{\text{true positive}}{\text{true positive + false negative}}$), which measure the proportion of accurate positive predictions and the ability

to find all the positive samples, respectively. As precision and recall are inversely related, we used their harmonic mean, known as the $F_1$ score $(2 \times \frac{precision \times recall}{precision + recall})$, as our evaluation metric to maximize both precision and recall simultaneously. This score helps maintain a balance between precision and recall, with a high score indicating a more accurate model (Figure S6). We then fitted the tuned model to the train and test sets. The log-odds polynomial degree with the highest F1 score on the test set was selected as the best degree. Lastly, we utilized the multiclass logistic regression with the chosen log-odds polynomial degree to fit the entire dataset (the recombined train and test set) and obtained an optimized phase diagram (Figure S7).

With the selected best model, we estimated the uncertainty in our predicted phase boundaries using bootstrapping. We resampled the compiled dataset with replacement (67%) and created $5 \times 10^3$ copies of the bootstrapped dataset. We then fit the best model to each resampled copy of the dataset, resulting in $5 \times 10^3$ sets of phase boundaries. We report 1–99% confidence intervals of the simulated distributions as the allowed $P$–$T$ regions (or uncertainties) for the phase boundaries (Fig. 2).

### A composite mantle temperature model for the "660-km" discontinuity

Our composite mantle temperature map for the "660-km" discontinuity provides an estimate of temperature variations at 660 km depth within the Earth's mantle, labeled as T660. It is a combination of three existing thermal models: 1) 2D kinematic models for subducting slabs from Syracuse et al. [35]; 2) global shear velocity constraints for plume-fed hotspots from Bao et al. [36]; and 3) globally-compiled S660S observations for the ambient mantle from Waszek et al. [57]. The values of T660 are estimated as follows:

- **Slabs:** To estimate slab temperatures at 660 km depth, we extrapolated the slab surface $P$–$T$ paths of all arc segments, using the W1300 case from [S10]. This process involved: 1) analyzing subduction temperature profiles, which become mostly constant after 150 km depth; 2) calculating the average slope of each temperature profile at 150–250 km depth; and 3) estimating T660 for each subduction segment by extrapolating their $P$–$T$ paths to 660 km depth using their respective temperature gradients.

- **Plumes:** For plume-fed oceanic hotspots, T660 estimates were based on mantle potential temperature estimates from [B22]. This process involved: 1) calculating mantle adiabats for each plume-fed hotspot using the HeFESTo code[30,58], assuming a depleted MORB mantle (DMM)[59]; and 2) extracting the mantle temperature at 660 km depth from each mantle adiabat.

- **Ambient mantle:** The same process was applied to the ambient mantle using the global mantle potential temperature map from [W21]. Here, we assumed a mechanically mixed (MM) mantle composed of 20% basalt and 80% harzburgite[57,59].

We note that our estimates for T660 could have significant errors, especially for the slabs, due to the simple extrapolation of the slab surface temperature with a constant temperature gradient beyond 250 km depth. This could lead to deviations from the actual mantle temperatures at 660 km depth for some slab segments[60]. The estimates for the slabs serve here only as an upper bound and will not represent the minimum temperature within the slabs, which some regional models suggest could be a few hundred K lower, e.g., in the western Pacific[61], and thus our estimates for $\gamma_{post-spinel}$ serve here as a lower bound. In addition, another bm-forming reaction, ak ↔ bm, may replace the post-spinel transition in a small number of extremely cold slabs where T660 is below 1400 K, such as the Tonga slab (T660 = ~1360 K). Such extremely cold conditions for the ak ↔ bm transition are unlikely to be prevalent in the ambient mantle today, but they may be more common than those identified in the slabs of Fig. 3, if the actual T660 is lower than our estimates.

Despite these potential errors, our composite T660 model provides a reasonable first-order approximation of the lateral variations in mantle temperature at the 660-km discontinuity. The mantle temperature distributions in these three independently-constrained thermal models are consistent with one another; for example, mantle temperature is lowest around the western Pacific subduction zones, while plume-fed hotspots coincide with positive T660 anomalies in the ambient mantle (Fig. 3e).

## Data availability
The source data used to reproduce all of the figures in this study can be accessed in the Supplementary Data 1–5 and are available on Zenodo at https://doi.org/10.5281/zenodo.14188625.

## Code availability
The code used to construct the $Mg_2SiO_4$ phase diagram in this study, including detailed documentation and a benchmark case, can be accessed in the Supplementary Data 6–7, and are available on Zenodo via https://doi.org/10.5281/zenodo.14188625.

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

## Acknowledgements
J.D. was supported by a James Mills Peirce Fellowship from the Graduate School of Arts and Sciences at Harvard University. R.F. was supported by her faculty start-up fund at Harvard University and by the Henry Luce Foundation. M.B. was supported by a National Science Foundation Graduate Research Fellowship (DGE-1745303). We thank Andrew J. Campbell for loaning us his short symmetric cell for the gas membrane experiments. The experimental part of this work was performed at GeoSoilEnviroCARS (Sector 13) and HPCAT (Sector 16), Advanced Photon Source (APS), Argonne National Laboratory. The Advanced Photon Source is a U.S. Department of Energy (DOE) Office of Science User Facility operated for the DOE Office of Science by Argonne National Laboratory under Contract No. DE-AC02.06CH11357. GeoSoilEnviroCARS is supported by the National Science Foundation (EAR–1634415) and DOE (DE-FG02-94ER14466). HPCAT operations are supported by the National Nuclear Security Administration (DOE-NNSA)'s Office of Experimental Sciences. The data analysis part of this work was inspired by a Harvard class, CS109a: Introduction to Data Science, taught by Pavlos Protopapas, Kevin A. Rader, and Chris Tanner in Fall 2020, and this work benefited from its course materials.

## Author contributions
J.D. and R.F. designed the study. J.D., R.F., M.B., K.D., T.S., and K.T. conducted the experimental work with support from beamline scientists Y.M. and V.P. Analysis and interpretation of experimental data were carried out by J.D., R.F., and L.S. Thermodynamic modeling was performed by J.D. The manuscript was written by J.D. with input from all authors.

## Competing interests
The authors declare no competing interests.
