## [Peer Review File · Nature Communications]

Nonlinearity of the post-spinel transition and its expression in slabs and plumes worldwide

Corresponding Author: Dr Junjie Dong

Version 1:

Reviewer comments:

Reviewer #1

(Remarks to the Author)

Review of the manuscript "Nonlinearity of the post-spinel transition and its expression in slabs and plumes worldwide" by Dong et al. to Nature Communications.

This is a revision of the manuscript that was previously submitted to Nature Geoscience. I appreciate the authors' efforts to improve the manuscript. However, the main issue remains unresolved. Consequently, I do not believe it is suitable for high-level journals such as Nature Communications.

I previously raised concerns about the motivation for the current study. The authors have added a paragraph to the introduction addressing this point. They claim that the experimental temperatures in previous studies on the Clapeyron slope were significantly lower than those in the mantle transition zone (1600 - 2200 K), thereby questioning the reliability of predicting the Clapeyron slope in the mantle. However, this assertion is quite misleading. As demonstrated in previous research, such as Chanyshv et al. (2022), the experimental temperature range actually spans from 1200 to 2100 K.

Regarding the thermodynamic model, I fully agree that classification models can provide reasonable (or sometimes even better) fittings compared to thermodynamic models. I also concur that existing thermodynamic models in the literature may not accurately predict the post-spinel transition slope. However, this is because some thermodynamic parameters have not been precisely determined in the literature. As the authors mentioned, the ΔS and ΔV values in the literature are unreliable, thus impairing the thermodynamic model's ability to describe the post-spinel phase transition.

Given this, a more reasonable strategy would be to obtain more precise thermodynamic parameters using the current data, rather than constructing a classification model, which lacks physical meaning. A classification model is appropriate only when the physical connections between the values are unknown. For the current case, we already know that the phase boundary is determined by thermodynamics. Therefore, the task for experimentalists should be to precisely determine the thermodynamic parameters, such as ΔS and ΔV .

As I mentioned in my previous review, the main problem with previous studies is the limited precision (or accuracy) in determining the P-T conditions of the phase boundaries. While the authors have added more data points, these additions do not offer an improvement over the previous studies. In fact, some of their data points, such as the orange symbols in the wd and rw stability fields, contradict their results. Consequently, their findings do not provide a better constraint on the Clapeyron slope compared to previous studies. The authors have discussed the difference between precision and accuracy, arguing that multi-anvil experiments may have systematic errors in temperature readings. This might be true, but following this logic, the authors should focus on obtaining more data points closer to the phase boundaries to better constrain them. While I agree that data points far from the phase boundaries can help to some extent, "direct observations" near the phase boundaries provide the most precise constraints for the phase boundaries. We can easily imagine that IF there are only data points far from the phase boundaries, we cannot precisely determine the phase boundaries. In contrast, data points (if they are correct) near the phase boundaries can directly determine these boundaries with much higher accuracy. This is the fundamental principle of mathematics.

Reviewer #2

(Remarks to the Author)

The authors have addressed all my concerns. It may be accepted for the publication in nature communications.

Reviewer #3

(Remarks to the Author)

I am satisfied with the discussion of the limitations of the presented temperature model T660. I support the publication of the manuscript.

Reviewer #4

(Remarks to the Author)

Dear Editor,

The Claperon slope at the 660 km discontinuity is an important question with significant geophysical implications. Ideally, the problem can be resolved by covering the relevant pressure-temperature space with high-density experimental points (many high quality experiments precision and high accuracy). However, high-temperature and high-pressure experiments are expensive and time-consuming, leaving the problem still unresolved. Dong et al. provide new experimental results in this study and use a statistical/machine learning approach to investigate the problem. The approach is promising and advances our understanding of the Claperon slope near 660 km depth, which might be non-linear.

The study provides statistical and machine learning evidence by:

- 1) Compiling and filtering the experimental results to construct a self-consistent database. This is important because although the accuracy of temperature is not good enough for diamond anvil cells, the dataset is self-consistent in revealing the Claperon slope.
- 2) A multi-class logistic regression is used, which is reasonable for the phase diagram's classification problem.

I evaluate the data, method, and results, in particular of the statistical and machine learning parts. Given the data in the supplementary Table, I organized the data and investigated the problem independently using Geochemistry π Python Framework version 0.6.1 (ZhangZhou and He et al. 2024 G Cubed). I tried several classification machine learning methods. The results suggest non-linear phase boundaries are more consistent with the available experiment data.

The results will generate broad interest in the experimental, geodynamic, and seismological fields. I recommend publication in Nature Communications after minor revision.

I have two suggestions:

- a) The authors must provide the codes and training/test data in Zenodo for a final review before acceptance. The code should open with the publication and be clearly annotated for the dataset and code. In my test exercise using Geochemistry π Python Framework, I have to work on the supplementary Table to get the training data.
- b) Support Vector Machines (SVM) are the classic tool for classification problems (Cortes and Vapnik, 1995) and, therefore, is usually the first algorithm considered for classification. With kernel tricks, SVM can also achieve non-linear boundaries. The authors can consider mentioning SVM in the manuscript to acknowledge people's work on classification problems using machine learning. I think it is redundant to present SVM, XGBoost results in the manuscript. Logistic regression can justify the non-linear phase boundary problem.

Dataset S5a: Plume-fed oceanic hotspots accompanied by the post-spinel transntion (transition, typo)

Fig.S1. I suggest putting the P-T phase boundary calculated by thermodynamic code HeFESTo (e.g., blue lines in Figure S1) as an equation here. The authors can compare the classification results of the equation (HeFESTo model) with compiled experimental data (confusion matrix results). Furthermore, the authors can compare machine learning models (different degrees of polynomials) with experimental data. The above results can be put in Dataset S6. I think this can further justify the advancement brought by machine learning.

References:

Cortes and Vapnik (1995) Support-vector networks. Machine learning, 20, 273–297. <https://doi.org/10.1007/BF00994018>

J ZhangZhou, Can He, Jianhao Sun, Jianming Zhao, Yang Lyu, Shengxin Wang, Wenyu Zhao, Anzhou Li, Xiaohui Ji, Anant Agarwal (2024) Geochemistry π : Automated machine learning python framework for tabular data. Geochemistry Geophysics Geosystems 25, e2023GC011324. <https://agupubs.onlinelibrary.wiley.com/doi/full/10.1029/2023GC011324>

Version 2:

Reviewer comments:

Reviewer #1

(Remarks to the Author)

Although I do not fully agree with the authors' points, I acknowledge that they have provided reasonable counterarguments. Therefore, I will not insist on rejecting it. I have no further comments.

(Remarks on code availability)

Reviewer #4

(Remarks to the Author)

The authors have answered my questions. Data and codes are both available to readers. I think the manuscript is good to go.

One suggestion:

Dataset-S1: In the read me document, you can define $t_err=temp-temp_ds$. Otherwise, it is confusing at a first glance to see the negative values of t_err in the worksheets in Dataset-S1.

(Remarks on code availability)

Yes. There are README files for both code and data, which are enough for installation and run the code. See attached supplementary material for the reproduction result.

REPLY TO REVIEWER #1

REVIEWER'S COMMENTS IN BLACK

AUTHORS' RESPONSE IN RED

MANUSCRIPT REVISION IN BLUE

Reviewer #1 (Remarks to the Author):

Review of the manuscript "Nonlinearity of the post-spinel transition and its expression in slabs and plumes worldwide" by Dong et al. to Nature Communications.

This is a revision of the manuscript that was previously submitted to Nature Geoscience. I appreciate the authors' efforts to improve the manuscript. However, the main issue remains unresolved. Consequently, I do not believe it is suitable for high-level journals such as Nature Communications.

I previously raised concerns about the motivation for the current study. The authors have added a paragraph to the introduction addressing this point. They claim that the experimental temperatures in previous studies on the Clapeyron slope were significantly lower than those in the mantle transition zone (1600 - 2200 K), thereby questioning the reliability of predicting the Clapeyron slope in the mantle. However, this assertion is quite misleading. As demonstrated in previous research, such as Chanyshv et al. (2022), the experimental temperature range actually spans from 1200 to 2100 K.

We apologize for any confusion on this point; we have now clarified in the main text (L59) that:

“Our new data span multiple phase boundaries in Mg_2SiO_4 , with a particular focus on the previously-understudied triple point of wd-rw-bm+pe. Combining our LH-DAC dataset including the triple point with previous high-precision multi-anvil (MA) press data, we used logistic regression and supervised learning methods to identify the phase boundaries in Mg_2SiO_4 under mantle transition zone conditions and determine the nonlinearity of the post-spinel boundary.”

The particular utility of our new higher temperature dataset (to ~2600 K, instead of ~2100 K in Chanyshv et al.) is that it significantly improves constraints on the location of the triple point at ~2200 K and thus the slopes and non-linearity of the phase

boundaries that meet there, in addition to our new constraints on the post-spinel phase boundary directly. While the Chanyshv et al. dataset does span much of the MTZ temperature range, the phase boundary slopes at MTZ temperatures are much better constrained with the inclusion of our new data spanning a wider range of temperatures.

We also added an additional SI figure to contrast the P-T coverage of our work (red) with that of previous work (blue). The location of the triple point is indicated by a yellow star to emphasize its critical role in understanding the nonlinearity of the post-spinel transition:

SI, adding Fig. S3:

Fig. S5. Comparison between the P - T ranges investigated in previous experiments and in this work. The red arrow and symbols represent the experimental P - T range covered in this study (1500 K to 2500 K), while the blue arrow and symbols indicate the P - T range covered by previous studies, including Chanyshv et al. (2022) (19), which is limited to 1500–2100 K. The yellow star marks the triple point derived from our analysis, which plays a critical role in constraining the shape and nonlinearity of the post-spinel boundary and is better determined with the addition of our new data to 2500 K.

Regarding the thermodynamic model, I fully agree that classification models can provide reasonable (or sometimes even better) fittings compared to thermodynamic models. I also concur that existing thermodynamic models in the literature may not accurately predict the post-spinel transition slope. However, this is because some thermodynamic parameters have not been precisely determined in the literature. As the authors mentioned, the ΔS and ΔV values in the literature are unreliable, thus impairing the thermodynamic model's ability to describe the post-spinel phase transition.

Given this, a more reasonable strategy would be to obtain more precise thermodynamic parameters using the current data, rather than constructing a classification model, which lacks physical meaning. A classification model is appropriate only when the physical connections between the values are unknown. For the current case, we already know that the phase boundary is determined by thermodynamics. Therefore, the task for experimentalists should be to precisely determine the thermodynamic parameters, such as ΔS and ΔV .

We appreciate your comments regarding measurements of thermodynamic parameters such as ΔS and ΔV , and we agree that having more accurate values for those parameters would be very helpful to determining the phase boundary and should be a goal for our community. However, to our knowledge, these parameters cannot be reliably derived from our dataset or any others on the relevant phases at the relevant conditions. For example, current experimental capabilities for calorimetric measurements of ΔS are not sufficient to resolve the nonlinearity of the post-spinel boundary. We have stated in the main text:

Main text lines 43-46:

“However, the empirical thermodynamic models rely on experimental data collected at low temperatures, such as heat capacity (<500 K)^{20, 21}, which, when extrapolated, produce inaccurate entropy and volume at the high temperatures

relevant to the mantle transition zone (1600–2200 K), thus compromising their reliability in predicting the Clapeyron slope.”

For this reason, we have chosen to rely instead on direct experimental observations of phase stability. The classification model serves as a complementary approach (to the thermodynamic model) to interpret phase relations over a broad temperature range, integrating both new and existing data. While complete thermodynamic information on all phases involved would indeed be ideal, our main goal is to estimate the nonlinearity of the post-spinel boundary and more importantly understand its implications for mantle dynamics. We consider the quantification of the post-spinel nonlinearity from the classification model adequate for the purpose, and our use of this model, combined with our larger dataset that extends to higher temperatures, provides significantly better constraints on the nonlinearity of this phase boundary than in any previous work

As I mentioned in my previous review, the main problem with previous studies is the limited precision (or accuracy) in determining the P-T conditions of the phase boundaries. While the authors have added more data points, these additions do not offer an improvement over the previous studies. In fact, some of their data points, such as the orange symbols in the wd and rw stability fields, contradict their results. Consequently, their findings do not provide a better constraint on the Clapeyron slope compared to previous studies. The authors have discussed the difference between precision and accuracy, arguing that multi-anvil experiments may have systematic errors in temperature readings. This might be true, but following this logic, the authors should focus on obtaining more data points closer to the phase boundaries to better constrain them. While I agree that data points far from the phase boundaries can help to some extent, "direct observations" near the phase boundaries provide the most precise constraints for the phase boundaries. We can easily imagine that IF there are only data points far from the phase boundaries, we cannot precisely determine the phase boundaries. In contrast, data points (if they are correct) near the phase boundaries can directly determine these boundaries with much higher accuracy. This is the fundamental principle of mathematics.

We agree that observations closer to the phase boundary provide the most direct constraints, and fully recognize their importance, especially the contributions from the in situ high-precision multi-anvil results. We also acknowledge that the laser-

heated diamond anvil cell (DAC) technique used in our study has larger random errors compared to large-volume press experiments under similar pressure-temperature conditions (but the relatively small number of data in Fig 2 in the wrong stability field are all within uncertainty (in P - T) of the phase boundaries, shown in Fig. S2). However, reaching the higher temperatures necessary to study the triple point is only possible with the LH-DAC, and our study specifically aims to 1) provide experimental constraints on the triple point as it provides critical information on the shape of the post-spinel boundary, and 2) combine the LH-DAC data including the triple point and the high-precision MA data at lower temperature to determine the post-spinel boundary. We have added a paragraph to the main text discussing this issue and our experimental limitations (L110–118):

“Our new LH-DAC dataset covers a wider temperature range that extends beyond the triple point of $wd-rw-(bm+pe)$, which is critical for constraining the nonlinearity of the post-spinel boundary and is only possible in the LH-DAC. However, using the LH-DAC to achieve these higher temperatures comes with the trade-off of lower precision (e.g., in P and T) than in a multi-anvil press¹⁹. Here, we combine our new LH-DAC data with the selected MA data from the literature, to take advantage of the higher-precision constraints on the phase boundary at lower temperatures obtained in those studies while significantly improving constraints on the triple point, and thus the nonlinearity of the post-spinel boundary, through the addition of our higher temperature data. Future work should aim to probe closer to the post-spinel boundary with improved experimental setups that balance achieving high temperatures and measurement precision.”

Despite this limitation, we believe that our current methodology, which combines new triple point data (between ~ 1500 and ~ 2700 K, by LH-DAC) with curated literature data (between ~ 1500 and ~ 2100 K, by MA) and advanced data analysis, provides a comprehensive constraint on the nonlinearity of the post-spinel transition that was not previously possible. We consider this work to be a significant step forward in understanding this complex but important transition in the mantle.

REPLY TO REVIEWER #2

REVIEWER'S COMMENTS IN BLACK

AUTHORS' RESPONSE IN **RED**

MANUSCRIPT REVISION IN **BLUE**

Reviewer #2 (Remarks to the Author):

The authors have addressed all my concerns. It may be accepted for the publication in nature communications.

Thank you for your support in the publication of the revised manuscript.

REPLY TO REVIEWER #3

REVIEWER'S COMMENTS IN BLACK

AUTHORS' RESPONSE IN **RED**

MANUSCRIPT REVISION IN **BLUE**

Reviewer #3 (Remarks to the Author):

I am satisfied with the discussion of the limitations of the presented temperature model T660. I support the publication of the manuscript.

Thank you for acknowledging that we have addressed your concerns about the 660-km temperature model. We appreciate your support in publishing the revised manuscript.

REPLY TO REVIEWER #4

REVIEWER'S COMMENTS IN BLACK

AUTHORS' RESPONSE IN RED

MANUSCRIPT REVISION IN BLUE

Reviewer #4 (Remarks to the Author):

Dear Editor,

The Claperon slope at the 660 km discontinuity is an important question with significant geophysical implications. Ideally, the problem can be resolved by covering the relevant pressure-temperature space with high-density experimental points (many high quality experiments precision and high accuracy). However, high-temperature and high-pressure experiments are expensive and time-consuming, leaving the problem still unresolved. Dong et al. provide new experimental results in this study and use a statistical/machine learning approach to investigate the problem. The approach is promising and advances our understanding of the Claperon slope near 660 km depth, which might be non-linear.

The study provides statistical and machine learning evidence by:

- 1) Compiling and filtering the experimental results to construct a self-consistent database. This is important because although the accuracy of temperature is not good enough for diamond anvil cells, the dataset is self-consistent in revealing the Claperon slope.
- 2) A multi-class logistic regression is used, which is reasonable for the phase diagram's classification problem.

I evaluate the data, method, and results, in particular of the statistical and machine learning parts. Given the data in the supplementary Table, I organized the data and investigated the problem independently using Geochemistry π Python Framework version 0.6.1 (ZhangZhou and He et al. 2024 G Cubed). I tried several classification machine learning methods. The results suggest non-linear phase boundaries are more consistent with the available experiment data.

The results will generate broad interest in the experimental, geodynamic, and seismological fields. I recommend publication in Nature Communications after minor revision.

Thank you for recognizing the novelty and importance of our work, and we are particularly grateful for your efforts to independently verify our conclusions. We appreciate the constructive feedback and address the suggested revisions as follows.

I have two suggestions:

a) The authors must provide the codes and training/test data in Zenodo for a final review before acceptance. The code should open with the publication and be clearly annotated for the dataset and code. In my test exercise using Geochemistry π Python Framework, I have to work on the supplementary Table to get the training data.

We apologize for any inconvenience caused by the lack of access to the source code. We have now organized and thoroughly annotated the code, which is included as a supplementary file in this resubmission ('dataset-s7.zip'). The code is designed for general use in phase diagram construction, with a benchmark example of Mg_2SiO_4 provided to reproduce the results presented in this manuscript. Auxiliary files, including a README.md with detailed instructions and the Python environment files (environment.yml and requirements.txt), are also provided to ensure ease of use and reproducibility. In addition, the code has been deposited in a GitHub repository, (MLPD: <https://github.com/dong2j/MLPD.git>) for wider access and archived on Zenodo for version control (MLPD: <https://doi.org/10.5281/zenodo.13988951>).

We have added the description and link of the source code in the main text and in the supplementary material of the manuscript:

Main Text Line 312–318, adding:

“Data and code availability statements

All data and codes used to construct the Mg_2SiO_4 phase diagram in this study, including detailed documentation, are available as Supplementary Information. For long-term accessibility and version control, the code has also been deposited in a GitHub repository (<https://github.com/dong2j/MLPD.git>) and archived on Zenodo (<https://doi.org/10.5281/zenodo.13988951>).

b) Support Vector Machines (SVM) are the classic tool for classification problems (Cortes and Vapnik, 1995) and, therefore, is usually the first algorithm considered for classification.

With kernel tricks, SVM can also achieve non-linear boundaries. The authors can consider mentioning SVM in the manuscript to acknowledge people's work on classification problems using machine learning. I think it is redundant to present SVM, XGBoost results in the manuscript. Logistic regression can justify the non-linear phase boundary problem.

We appreciate your additional context on classification algorithms. We have revised the text to acknowledge methods such as SVM and XGBoost and to direct readers to general ML resources such as scikit-learn as well as geochemistry-specific ML tools such as Geochemistry π :

SI Lines 172–180, adding:

“To construct the phase diagram, we used logistic regression, which is a well-suited model for this study due to its simplicity and interpretability. While more complex classification algorithms such as Support Vector Machines (SVMs) and XGBoost are able to model nonlinear boundaries using kernel functions or decision tree ensembles, they often produce unrealistic oscillations in the phase boundaries. This behavior is particularly problematic when working with limited or noisy experimental data, as these classifiers often overfit the data to produce frequent sign reversals in the phase boundary slope---such behavior contradicts the thermodynamic expectation for solid-solid phase transitions. In contrast, the result of the logistic classifier is more physically consistent with the gradual, non-reversing changes in slope characteristic of solid-solid phase transitions. Details on the basics and differences of several common classification algorithms can be found in general machine learning resources such as “scikit-learn” (17) as well as geochemistry-specific machine learning resources such as “Geochemistry π ” (18).”

We also agree that it is redundant to present results from SVMs and XGBoost for this study. We chose logistic regression for the following reasons:

- 1. Interpretability: SVMs use kernel functions to transform data, while XGBoost relies on ensembles of decision trees, both of which make interpretation difficult. In contrast, logistic regression is mathematically simple, and its functional form matches well the physical process expected in first-order phase transitions.**

- 2. Thermodynamic consistency: Our thermodynamic understanding suggests that solid-solid phase boundaries should be nonlinear over a broad P-T space, with a gradual change in slope that does not exhibit abrupt sign reversals. However, complex models such as XGBoost and SVMs can produce boundaries with unrealistic swings, especially when data are limited or noisy (as in this high-pressure experimental context). The simplicity of logistic regression minimizes overfitting, making it more suitable for this study.**

Dataset S5a: Plume-fed oceanic hotspots accompanied by the post-spinel transition (transition, typo)

We have fixed the typo in Dataset S5.

Fig.S1. I suggest putting the P-T phase boundary calculated by thermodynamic code HeFESTo (e.g., blue lines in Figure S1) as an equation here. The authors can compare the classification results of the equation (HeFESTo model) with compiled experimental data (confusion matrix results). Furthermore, the authors can compare machine learning models (different degrees of polynomials) with experimental data. The above results can be put in Dataset S6. I think this can further justify the advancement brought by machine learning.

Thank you for your suggestion to add the confusion matrices to demonstrate and further justify the progress made by machine learning. We have now included a comparison of the confusion matrices between the logistic model and the thermodynamic code HeFESTo as SI Fig. S8 instead of adding the results to dataset S6. As expected, the logistic inversion of the experimental data yields a higher classification accuracy compared to the thermodynamic model of the post-spinel transition of HeFESTo.

SI Lines: 48–49, adding:

“We also examined the confusion matrices and found that our logistic inversion of direct experimental observations provides higher classification accuracy compared to the HeFESTo prediction at the post-spinel boundary. (Figure S8)”

adding Figure S8 with a caption:

“

Fig. S8. Comparison of normalized confusion matrices for phase diagram models using (a) the logistic classifier (this work) and (b) the thermodynamic model (HeFESTo), evaluated for the post-spinel boundary between rw and bm+pe. The diagonal elements represent correctly predicted phases, while the off-diagonal elements indicate misclassified phases. The logistic classifier achieves higher scores for both phases, with an accuracy of 79% for rw and 90% for bm+pe, compared to 74% and 89% for HeFESTo, respectively.”

References:

Cortes and Vapnik (1995) Support-vector networks. Machine learning, 20, 273–297. <https://doi.org/10.1007/BF00994018>

J ZhangZhou, Can He, Jianhao Sun, Jianming Zhao, Yang Lyu, Shengxin Wang, Wenyu Zhao, Anzhou Li, Xiaohui Ji, Anant Agarwal (2024) Geochemistry π : Automated machine learning

REPLY TO REVIEWER #4

python framework for tabular data. *Geochemistry Geophysics Geosystems* 25,
e2023GC011324. <https://agupubs.onlinelibrary.wiley.com/doi/full/10.1029/2023GC011324>

REPLY TO REVIEWER

REVIEWER'S COMMENTS IN BLACK

AUTHORS' RESPONSE IN **RED**

Reviewer #1 (Remarks to the Author):

Although I do not fully agree with the authors' points, I acknowledge that they have provided reasonable counterarguments. Therefore, I will not insist on rejecting it. I have no further comments.

We appreciate your input and thank you for acknowledging our reasoning that supports the conclusion!

Reviewer #4 (Remarks to the Author):

The authors have answered my questions. Data and codes are both available to readers. I think the manuscript is good to go.

One suggestion:

Dataset-S1: In the read me document, you can define $t_{err} = temp - temp_{ds}$. Otherwise, it is confusing at a first glance to see the negative values of t_{err} in the worksheets in Dataset-S1.

We fixed the missing definition in the README file.

Reviewer #4 (Remarks on code availability):

Yes. There are README files for both code and data, which are enough for installation and run the code. See attached supplementary material for the reproduction result.

We appreciate your efforts to replicate our work and your support for publication!